# Persistently Worsened Tear Break-up Time and Keratitis in Unilateral Pseudophakic Eyes after a Long Postoperative Period

**DOI:** 10.3390/biomedicines8040077

**Published:** 2020-04-05

**Authors:** Akiko Hanyuda, Kazuno Negishi, Kazuo Tsubota, Masahiko Ayaki

**Affiliations:** 1Department of Ophthalmology, Keio University School of Medicine, Tokyo 160-8582, Japan; akikohanyuda@gmail.com (A.H.); fwic7788@mb.infoweb.ne.jp (K.N.); tsubota@z3.keio.jp (K.T.); 2Epidemiology and Prevention Group, Center for Public Health Sciences, National Cancer Center, Tokyo 104-0045, Japan; 3Department of Nutrition, Harvard T. H. Chan School of Public Health, Boston, MA 02215, USA; 4Otake Clinic Moon View Eye Center, Kanagawa 242-0001, Japan

**Keywords:** dry eye disease, cataract surgery, ocular surface distress, tear film instability

## Abstract

Dry eye disease may develop and persist after cataract surgery; however, unilateral cases have not been fully documented. This cross-sectional, observational study was conducted in five eye clinics in Japan. A total of 1023 outpatients were initially enrolled, and 89 unilateral pseudophakic subjects with 1+ year of follow-up after uncomplicated cataract surgery were included. The tear break-up times (TBUTs) and keratoconjunctival staining results were compared between phakic and pseudophakic eyes. The mean age of the patients was 69.3 ± 10.4 years (32 men, 36.0%), and the mean postoperative period was 4.6 ± 4.4 (1–20) years. For the ophthalmic parameters, the TBUTs were 4.4 ± 1.9 and 3.8 ± 1.9 s (*p* < 0.001), the keratoconjunctival staining scores were 0.11 ± 0.38 and 0.22 ± 0.56 (*p* = 0.02), the spherical equivalents were −1.27 ± 2.51 and −0.99 ± 1.45 D (*p* = 0.21), the astigmatic errors were 0.79 ± 0.66 and 0.78 ± 0.58 D (*p* = 0.80), and the intraocular pressures were 13.6 ± 2.9 and 13.5 ± 2.6 mmHg (*p* = 0.62) for the phakic and pseudophakic eyes, respectively. The corneal status was significantly worse in the pseudophakic eyes than in the contralateral phakic eyes, even after more than one year after implant surgery. The present results suggested that long-term ocular surface problems should be examined further since they may not originate only from surgery or postoperative ocular surface diseases.

## 1. Introduction

Dry eye disease (DED) is a multifactorial disease of the tears and ocular surface that results in symptoms of ocular discomfort, visual disturbance, and tear film instability [1,2], leading to a substantial decrease in job efficiency and quality of life [3]. Although numerous risk factors for DED, including an older age, the female sex, and an Asian background, have been identified [2], a wide variety of ocular interventions can cause DED [4]. In addition to the aging of the global population and a marked increase in the number of cataract surgeries performed, postoperative DED has become a significant public health concern for elderly adults, as these individuals have a higher incidence of pre-existing DED than younger adults [5].

Given that postoperative DED has been directly linked to patient dissatisfaction, visual disturbance, and poor surgical outcomes, the risk of developing DED after cataract surgery has been increasingly recognized [6,7,8,9,10,11,12,13]. According to a recent report from the Tear Film and Ocular Surface Society International Dry Eye Workshop (TFOS DEWS) II iatrogenic report [4], DED that develops after cataract surgery has been classified as “surgically induced iatrogenic DED”. In this report, ocular surface management is recommended perioperatively for all patients, even in the absence of preoperative DED.

The majority of the previous studies have shown a substantial decrease in the tear break-up time (TBUT) and increased corneal staining approximately 3 months after cataract surgery [6,14,15,16]. However, the long-term influence of postoperative DED remains unclear. As we previously reported [17], a specific population of pseudophakic patients presented with chronic ocular surface disturbance postoperatively, even after more than one year. A growing number of studies have also reported that a persistent decrease in the density of the conjunctival goblet cells [6,7,18] and meibomian gland dysfunction (MGD) without structural changes [19] occur following uncomplicated phacoemulsification.

Hence, the purpose of this study was to investigate the chronic influence of cataract surgery on the ocular surface in unilateral pseudophakic patients. The novelty of this study was related to the inclusion of subjects who underwent cataract surgery at least 1 year prior to study enrollment to minimize the direct influence of surgical procedures and transient corneal hyper/hypoesthesia that can occur during the wound healing process. We compared the TBUTs and keratoconjunctival staining results between eyes with an implanted intraocular lens (IOL) and those with a phakic lens. In the exploratory analyses, we further examined whether persistent ocular surface disturbance related to cataract surgery differed by the patient’s age or sex or the postoperative duration.

## 2. Materials and Methods 

### 2.1. Study Design, Ethical Approval, and Study Population

This multisite, hospital-based, cross-sectional study was conducted in five eye clinics, including those in Komoro Kosei General Hospital (Nagano, Japan), Shinseikai Toyama Hospital (Toyama, Japan), Tsukuba Central Hospital (Ibaraki), Todoroki Eye Clinic (Tokyo, Japan), and Otake Clinic Moon View Eye Center (Kanagawa, Japan). We initially enrolled 1023 eye care visitors who underwent IOL implantation from April 2015 to November 2019. This study followed the tenets of the 1995 Declaration of Helsinki (as revised in Edinburgh, 2000), and all the participants provided written informed consent. The respective institutional review boards and ethics committees of Shinseikai Toyama Hospital (permit number: 150503), Komoro Kosei General Hospital (permit number: 2705), and Kanagawa Medical Association (date of approval: 12 November 2018) approved this study.

### 2.2. Inclusion and Exclusion Criteria

In this study, we included 89 subjects (89 eyes with pseudophakia and 89 eyes with phakia) who had undergone cataract surgery unilaterally at least 1 year before the recruitment date and had a best corrected visual acuity of at least 20/25 bilaterally and no sign of capsular opacification in the operated eye. We included only subjects who had age-related cataracts and did not have any perioperative complications. The exclusion criteria were those who had incomplete data on the subjective dry eye symptoms or ocular surface parameters, including the keratoconjunctival staining scores and TBUT, and those with a treatment history of severe DED beyond the use of hyaluronic acid, a history of ocular diseases (i.e., glaucoma, age-related macular disease, or diabetic retinopathy), or ocular surgeries other than unilateral cataract surgery. In addition, because the main purpose of our current study was to evaluate dry eye signs and symptoms after cataract surgery, DED cases accompanied by severe conjunctivochalasis, superior limbic keratocojunctivitis, lid-wiper epitheliopathy, and filamentary keratitis, considered to be explained unrelated to surgical procedures, were excluded from the analyses.

### 2.3. Experimental Protocol

After identifying the participants who met the inclusion criteria above, we reviewed the participants’ medical records regarding the basic demographic characteristics and cataract surgical history. In all cases, cataract extraction was performed by skilled surgeons with the standard phacoemulsification and aspiration technique, and a posterior chamber IOL was implanted under topical anesthesia. Following a 2.75 mm corneal or corneoscleral incision, a side port approximately 1 mm in size was created 90 degrees away from the main incision, and a foldable IOL was successfully implanted in the capsular bag without major complications. In accordance with the current guidelines [20], all patients were prescribed nonsteroidal anti-inflammatory drugs (NSAIDs, 0.1% bromfenac or 0.1% diclofenac), topical steroids (0.1% betamethasone), and antibiotics (levofloxacin or moxifloxacin) for the first four postoperative weeks and only NSAIDs for the subsequent two months.

### 2.4. Ophthalmological Examinations and Interviews for Subjective Symptoms

Standardized ocular examinations were performed by trained ophthalmologists. The intraocular pressurewas determined by 3 successive readings with a noncontact tonometer (TonorefTM II, Nidek Co., Ltd., Aichi, Japan). The presence of subjective dry eye symptoms (i.e., dry sensation, foreign-body sensation, ocular pain, ocular fatigue, sensitivity to bright light, and blurred vision) was determined by the self-reported questionnaires. According to the standardized DED evaluation, [21] the fluorescein staining of tears was strictly monitored with no change of the subject’s aqueous tear volume, after putting 2 drops of saline solution with a fluorescein test strip (Showa Yakuhin Kako Co., Tokyo, Japan). The strip was touched gently to the central lower lid margin. We asked the patient to close the eye gently and briskly open the eye after several natural blinks. The investigator determined the starting point of eye opening as well as confirm the reproducibility of TBUT by 3 successive observations and the mean value was used for the current analyses. To minimize interexaminer heterogeneity, objective dry eye signs, including the mean TBUT and keratoconjunctival staining scores (0–9 points) based on the Japanese dry eye diagnostic criteria [22], were all evaluated by a single dry eye specialist (M.A.). The corneal and conjunctival fluorescein staining scores were evaluated in three areas (the temporal bulbar conjunctiva, nasal bulbar conjunctiva, and cornea) and scored on a 0–3-point scale in each section (0: no damage, 3: damaged entirely). Then, each score was summed, and the maximum score possible was a total of 9 points [22]. All examinations were performed at a temperature of 18–25 °C with humidity of 40%–60%.

### 2.5. Statistical Analysis

To assess the baseline characteristics of the study participants, the mean values for the continuous variables and proportions for the categorical variables were documented. To compare the ocular surface status characterized by the keratoconjunctival staining scores and TBUTs, paired t-tests were conducted according to the patients’ cataract surgical histories. In secondary analyses, we conducted stratification analyses and examined whether the occurrence of ocular surface disturbance after cataract surgery differed by the patient’s age (<71 vs. ≥71 years old; cut-off point for the median age in the study participants), sex (men vs. women), or postoperative duration (<3 vs. ≥3 years; cut-off point for the median postoperative years). To further explore the risk factors related to ocular surface abnormalities, Spearman correlation analyses across the demographic factors and between the phakic and pseudophakic eyes were conducted. In addition, odds ratios for developing tear instability (TBUT ≤5 seconds) for each risk factor were compared between the phakic and pseudophakic eyes. All statistical tests were two-sided, and the significance level was set to an α of 0.05. All analyses were performed using SAS software (Version 9.4, SAS Institute, Cary, NC, USA).

## 3. Results

### 3.1. Baseline Characteristics of the Study Participants

Eighty-nine individuals who underwent unilateral cataract surgery were enrolled in this study. Table 1 shows the baseline demographic and clinical characteristics. The mean postoperative duration was 4.6 ± 4.4 years (1–20 years). The mean age was 69.3 ± 10.4 years. We had more women than men in this study. Among the subjective dry eye symptoms, ocular fatigue (21.4%) and ocular pain (2.3%) were the most and least common complaints, respectively.

### 3.2. Clinical Characteristics in Relation to Cataract Surgery

Table 2 shows the clinical characteristics in relation to cataract surgery. For the clinical parameters, the TBUTs were 4.4 ± 1.9 and 3.8 ± 1.9 seconds (*p* < 0.001), the keratoconjunctival staining scores were 0.11 ± 0.4 and 0.22 ± 0.6 (*p* = 0.02), the spherical equivalents were −1.27 ± 2.51 and −0.99 ± 1.45 D (*p* = 0.21), the astigmatic errors were 0.79 ± 0.66 and 0.78 ± 0.58 D (*p* = 0.80), and the intraocular pressures were 13.6 ± 2.9 and 13.5 ± 2.6 mmHg (*p* = 0.62) for the phakic and pseudophakic lenses, respectively. The pseudophakic lenses presented significantly worse dry eye signs than did the phakic lenses. There was no difference in the refraction or intraocular pressure between the eyes that did or did not undergo cataract surgery.

### 3.3. Stratified Analysis by Age, Sex, and Postoperative Duration after Unilateral Cataract Surgery

A stratified analysis by age, sex, and postoperative duration after unilateral cataract surgery was conducted (Table 3). Regardless of the patient’s age, ocular surface disturbance was more prevalent in pseudophakic lenses than in phakic lenses. Specifically, the difference in the TBUT was marked in patients aged ≥71 years (TBUT for phakic lens and IOL: 4.6 ± 1.9 and 3.8 ± 2.0 seconds, respectively, *p* = 0.005), but the mean value for each parameter was generally consistent between the <71- and ≥71-years-old age groups. Similarly, in both sexes, the dry eye signs were worse in the pseudophakic lenses than in the phakic lenses (men: TBUT for the phakic lens and IOL: 5.0 ± 1.4 and 4.4 ± 1.8 s, respectively, *p* = 0.01; women: 4.1 ± 2.0 and 3.4 ± 1.9 seconds, respectively, *p* = 0.01), although the difference in the keratoconjunctival staining score did not reach a statistically significant level (men: 0.06; women: 0.13). Of note, the TBUT was substantially decreased in the pseudophakic eyes after fewer than 3 years postoperatively compared with the phakic eyes (TBUT for the phakic lens and IOL: 4.6 ± 1.6 and 3.7 ± 1.9 s, respectively, *p* = 0.003).

### 3.4. Risk Factors for Postoperative DED

The Spearman correlation analyses revealed that an older age, the male sex, and a short postoperative duration led individuals to be more susceptible to developing postoperative DED (i.e., shorter TBUTs or higher keratoconjunctival staining scores), although the statistical power was limited (Table 4). Similarly, men presented a significantly higher prevalence of tear instability than women, irrespective of whether the lens was phakic or pseudophakic (Table 5).

## 4. Discussion

We evaluated cases of persistent ocular surface disturbance, which had decreased TBUTs and increased keratoconjunctival staining scores, one year or more after cataract surgery in unilateral pseudophakic patients. In a sample of 89 patients who underwent unilateral IOL implantation, the pseudophakic eyes showed significantly worse clinical signs of dry eye than did the phakic eyes. Additional stratification analyses confirmed a significant tear film instability in the operated eyes, regardless of the patient’s age and sex and the duration of postoperative years. To the best of our knowledge, this study was the first to evaluate the long-term (at least 1 year after cataract surgery) influence of ocular surface disturbance in unilateral pseudophakic patients.

A growing body of evidence suggests that it is important to manage the ocular surface perioperatively in patients [6,7,8,9,10,11,12,13]. However, little is known about the persistent influence of ocular surface disturbance since most patients, especially after successful bilateral cataract extractions, were followed for at most several months. Thus, we focused on unilateral pseudophakic patients who were consistently followed for contralateral cataracts in our clinics. Of note, we found a significant decrease in the TBUT and increased keratoconjunctival staining in the pseudophakic eye compared with the phakic eye. These findings were generally consistent with those in previous studies with 1–12 months of follow-up after surgery [6,14,15,16]. Our current findings suggest that ocular surface abnormalities persist for more than one year after uncomplicated cataract surgery.

The pathophysiology of postsurgical DED varies across patients, and it includes responses to topical anesthetics, desiccation, light toxicity from operative microscopes, corneal nerve transection, and elevated inflammatory factors, which are generally no longer present a few months postoperatively [11]. Studies have also reported that corneal denervation induced by cataract surgery recovered to the baseline level within 1–3 months [7], which is unlikely to explain our current findings. Nonetheless, Han et al. suggested that meibomian gland function was persistently altered after cataract surgery; the use of meibography showed a marked decrease in meibum expressibility without accompanying structural changes in pseudophakic patients who were free from MGD preoperatively [19]. Choi et al. also suggested that a high Ocular Surface Disease Index (OSDI) at baseline and a short TBUT and MGD at 1 month postoperatively were risk factors for persistent DED after cataract surgery [23]. Other reports also suggested that conjunctival goblet cells and associated conjunctival cell squamous metaplasia had not recovered at 3 months after cataract surgery [6,7,18], further suggesting that managing the ocular surface in the relatively long term period is important.

Another possible explanation for the presence of persistent corneal damage after cataract surgery may be involved altered corneal mechanosensitivity mediating intrinsically photosensitive retinal ganglion cells (ipRGCs) [24,25]. With the existence of melanopsin, ipRGCs respond to light in the absence of rod and cone photoreceptor input. [26] IpRGCs project in the ciliary marginal zone [27] and iris [28], mediating innate light aversion into tissues in the anterior segments that receive trigeminal innervation [25]. Although melanopsin absorbs blue light with a peak absorption of approximately 460 to 480 nm [29,30], previous studies have reported a significant increase in blue light transmission and subsequent ipRGC activation in both blue-light blocking IOLs and clear IOLs after cataract surgery [31]. Considering these results and our current findings, we speculate that increased photoreception in the pseudophakia may activate the neural circuits that mediate corneal sensitivity and ocular surface disturbance, although additional studies are warranted.

This study has several limitations. First, due to the uniqueness of our study design, we had a small number of subjects who met the inclusion criteria and thus future studies in a large sample size were required to confirm the current findings. Second, the information on IOL materials was limited, although over 90% of our patients had yellow-tinted, hydrophobic acrylic IOLs (blue-light blocking IOLs). Third, the corneal sensitivity, lid hygiene status, and meibomian gland function were not measured objectively. We acknowledge this as a fundamental limitation since we could not assess whether the postoperative DED was attributed to the pathologic changes following cataract surgery, including persistent corneal nerve damage and MGD or other relevant factors, including poor hygiene of the operated eye (e.g., patients are likely to be afraid of washing around the operated eye) or prolonged use of postsurgical medications. Hence, additional studies with detailed ophthalmic evaluations, including those on corneal sensitivity and meibography, are required. In addition, this study was conducted in a cross-sectional manner; thus, we failed to infer the causality of our findings, although we examined all the medical records and confirmed that there were no substantial differences between the unoperated and operated eyes perioperatively. We could not eliminate selection bias (e.g., unilateral pseudophakic patients who did not undergo cataract surgery on the other eye were likely to be nervous); thus, the results should be interpreted with caution. Last, our study population comprised only Japanese patients; thus, future studies should be conducted in different populations.

Despite these drawbacks, our study has several strengths. Direct comparisons of pseudophakic and phakic eyes with a long follow-up period, with a median duration of 4.6 years, provided new insight into persistent postsurgical DED after cataract surgery. Furthermore, the ocular surface status was evaluated by a single experienced dry eye specialist (M.A.) in accordance with the most frequently used and standardized Japanese dry eye criteria, [22] which minimized inter-investigator heterogeneity.

In conclusion, postoperative DED characterized by a short TBUT and keratoconjunctivitis persisted more than one year after cataract surgery in unilateral pseudophakic patients. Although the underlying cause remains unclear, ophthalmologists should be aware of the need to manage the ocular surface postoperatively in the relatively long term.

## Figures and Tables

**Table 1 biomedicines-08-00077-t001:** Baseline demographic and clinical features of the study participants.

Baseline Characteristics	All (*n* = 89)
Mean age in years (SD)	69.3 (10.4)
Sex, *n* (%)	
Men	32 (36.0)
Women	57 (64.0)
Eye drop users for dry eye symptoms, *n* (%)	4 (5.0)
Mean years after cataract surgery (SD)	4.6 (4.4)
**Subjective Symptoms**	
Dry sensation, *n* (%)	10 (11.2)
Foreign-body sensation, *n* (%)	12 (13.5)
Ocular pain, *n* (%)	2 (2.3)
Ocular fatigue, *n* (%)	19 (21.4)
Sensitivity to bright light, *n* (%)	11 (12.4)
Blurred vision, *n* (%)	17 (19.1)
**Clinical Features**	
Mean tear break-up time, seconds (SD)	4.1 (1.7)
Mean keratoconjunctival staining score (SD)	0.24 (0.5)
Maximum blinking interval, sec (SD)	13.2 (3.3)
Mean spherical equivalent, D (SD)	−1.12 (1.80)
Mean astigmatism, D (SD)	0.78 (0.46)
Mean intraocular pressure, mmHg (SD)	13.6 (2.5)

SD, standard deviation.

**Table 2 biomedicines-08-00077-t002:** Ocular surface features in relation to cataract surgery.

Characteristics	Phakic Lens (*n* = 89)	IOL (*n* = 89)	*p* Value ^†^
Tear break-up time, seconds (SD)	4.4 (1.9)	3.8 (1.9)	<0.001
Keratoconjunctival staining score, (SD)	0.11 (0.38)	0.22 (0.56)	0.02
Spherical equivalent, D (SD)	−1.27 (2.51)	−0.99 (1.45)	0.21
Astigmatism, D (SD)	0.79 (0.66)	0.78 (0.58)	0.80
Intraocular pressure, mmHg (SD)	13.6 (2.9)	13.5 (2.6)	0.62

**^†^** Paired *t* test. IOL, intraocular lens; SD, standard deviation.

**Table 3 biomedicines-08-00077-t003:** Ocular surface features in relation to cataract surgery stratified by age, sex, and postoperative duration *.

Characteristics	Phakic Lens	IOL	*p* Value ^†^
**Age**			
**<71 years old (*n* = 42)**			
Tear break-up time, seconds (SD)	4.2 (1.8)	3.7 (1.8)	0.07
Keratoconjunctival staining score, (SD)	0.12 (0.5)	0.26 (0.7)	0.06
**≥71 years old (*n* = 47)**			
Tear break-up time, seconds (SD)	4.6 (1.9)	3.8 (2.0)	0.005
Keratoconjunctival staining score, (SD)	0.11 (0.3)	0.19 (0.5)	0.16
**Sex**			
**Men (*n* = 32)**			
Tear break-up time, seconds (SD)	5.0 (1.4)	4.4 (1.8)	0.01
Keratoconjunctival staining score, (SD)	0.06 (0.4)	0.22 (0.6)	0.06
**Women (*n* = 57)**			
Tear break-up time, seconds (SD)	4.1 (2.0)	3.4 (1.9)	0.01
Keratoconjunctival staining score, (SD)	0.14 (0.4)	0.23 (0.6)	0.13
**Postoperative duration**			
**<3 years after surgery (*n* = 40)**			
Tear break-up time, seconds (SD)	4.6 (1.6)	3.7 (1.9)	0.003
Keratoconjunctival staining score, (SD)	0.18 (0.5)	0.33 (0.7)	0.08
**≥3 years after surgery (*n* = 49)**			
Tear break-up time, seconds (SD)	4.3 (2.0)	3.8 (1.9)	0.07
Keratoconjunctival staining score, (SD)	0.06 (0.3)	0.14 (0.5)	0.10

***** Cut-off point for age and postoperative duration were the median value. **^†^** Paired *t* test. IOL. Intraocular lens; SD, standard deviation.

**Table 4 biomedicines-08-00077-t004:** Correlation between ocular surface features and patient demographics.

Characteristics	Tear Break-Up Time β ^†^ (*p*-Value)	Keratoconjunctival Staining Score β ^†^ (*p*-Value)
**Phakic Lens (*n* = 86)**		
Age	0.10 (0.35)	0.001 (0.99)
Sex (reference: women)	0.22 (0.04)	−0.15 (0.17)
Spherical equivalent	−0.06 (0.55)	0.01 (0.93)
**IOL (*n* = 86)**		
Age	0.01 (0.95)	−0.009 (0.94)
Sex (reference: women)	0.23 (0.03)	−0.004 (0.97)
Spherical equivalent	−0.12 (0.25)	0.14 (0.21)
Postoperative duration	0.07 (0.49)	−0.21 (0.05)

**^†^** Spearman correlation coefficient.

**Table 5 biomedicines-08-00077-t005:** Comparison of odds ratios for risk factors for developing tear instability ^†^.

Characteristics	OR (95% CI)	*p-*Value
**Phakic Lens (*n* = 86)**		
Age (continuous)	1.03 (0.99–1.07)	0.18
Sex (reference: women)	2.28 (0.93–5.58)	0.07
Spherical equivalent (continuous)	1.000 (0.998–1.001)	0.79
**IOL (*n* = 86)**		
Age (continuous)	1.02 (0.97–1.06)	0.48
Sex (reference: women)	2.90 (1.18–7.17)	0.02
Spherical equivalent (continuous)	0.997 (0.994–1.000)	0.05
Postoperative duration (continuous)	1.07 (0.97–1.18)	0.17

^†^ Tear instability was defined as tear break-up time <5 seconds.

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
