# Peer review of "Persistently Worsened Tear Break-up Time and Keratitis in Unilateral Pseudophakic Eyes after a Long Postoperative Period"

_biomedicines, 2020, doi:10.3390/biomedicines8040077_

Round 1

Reviewer 1 Report

I read with great interest the article by Hanyuda et al., attempting to evaluate the chronic influence of cataract surgery on the ocular surface in unilateral pseudophakic patients.Although the results of the work are interesting, several flaws and questions are raised in this manuscript that need to be addressed and clarified.

  1. The rationale of this study is rather vague. Dry eye is a generally bilateral problem mainly due to the 'reflex arc'. So, the cause of pseudophakic eye changes could be due to the contralateral eye (and vice versa). It seems inconclusive. Furthermore, visual discomfort could be related to the capsular opacification.
  2. Sample size calculation is fundamental for such a research.
  3. The Method is poor reliable/standardized. Did you measure the repeatability of results? Did you quantify the instilled fluorescein? Did you monitor the environmental conditions?

The article is interesting, but the interpretation may be incorrect. In my opinion, the paper at the moment is not ready for publication.

Author Response

Hanyuda A et al.  biomedicines-711379R

Response to Reviewer 1 Comments

Thank you very much for your interest in our manuscript entitled “Persistently worsened tear break-up time and keratitis in unilateral pseudophakic eyes after a long postoperative period”. To aid in the re-review of this manuscript, we have included a point-by-point response to each comment. The reviewer’s comments are italicized and placed in square brackets. In addition, within the revised manuscript, we have used underlined text to highlight changes in response to the reviewers’ comments.

We appreciate the suggestions and comments by the reviewer. As a consequence of valuable suggestions, we believe that our manuscript has been much improved.

[Reviewer 1:  I read with great interest the article by Hanyuda et al., attempting to evaluate the chronic influence of cataract surgery on the ocular surface in unilateral pseudophakic patients.Although the results of the work are interesting, several flaws and questions are raised in this manuscript that need to be addressed and clarified.]

We appreciate the reviewer’s complimentary comments. To improve our manuscript, we have addressed the issues as follows.

[Point 1.           The rationale of this study is rather vague. Dry eye is a generally bilateral problem mainly due to the 'reflex arc'. So, the cause of pseudophakic eye changes could be due to the contralateral eye (and vice versa). It seems inconclusive. Furthermore, visual discomfort could be related to the capsular opacification.]

We appreciate the reviewer’s comments. As the reviewer suggested, friction-related conditions including lid wiper epitheliopathy, conjunctivochalasis, and superior limbic keratoconjunctivitis were one cause of dry eye disease (DED) (Kawashima et al. Adv Ther. 2017; Viet Vu CH, et al. Ophthalmology. 2018); hence, we have excluded those who had such abnormalities in either eye in order to minimize the raised concern by the reviewer, suggesting that DED was induced by the “reflex arc” in the contralateral eye. The pathophysiology of DED is widely acknowledged as multifactorial (Stapleton F, et al. Ocul. Surf. 2017) and the “surgically induced iatrogenic DED”, which we currently evaluated, is linked to the direct and indirect surgical intervention including postoperative inflammation, corneal denervation, and meibomian gland dysfunction in the operated eye (Oh T, et al. Jpn.J.Ophthalmol. 2012; Sutu C, et al. Curr. Opin. Ophthalmol. 2016). Presumably, the severity of this type of DED was differed between the unoperated and operated eye (Gomes JAP, et al. Ocul. Surf. 2017).

In addition, all DED cases in this study were evaluated by a dry eye specialist (M.A.) and we have only included those with no sign of capsular opacification under slit lamp biomicroscopy or post-YAG laser capsulotomy. Further, the study participants had a best corrected visual acuity of at least 20/25 bilaterally. Conceivably, visual discomfort of the operated eye was unlikely related to the capsular opacification. In response to the reviewer’s comment, we have modified the METHODS section as follows:

“In this study, we included 89 subjects (89 eyes with pseudophakia and 89 eyes with phakia) who had undergone cataract surgery unilaterally at least 1 year before the recruitment date and had a best corrected visual acuity of at least 20/25 bilaterally and no sign of capsular opacification in the operated eye.

In addition, because the main purpose of our current study was to evaluate dry eye signs and symptoms after cataract surgery, DED cases accompanied by severe conjunctivochalasis, superior limbic keratocojunctivitis, lid-wiper epitheliopathy, and filamentary keratitis, considered to be explained unrelated to surgical procedures, were excluded from the analyses.”

(MATERIALS AND METHODS, page 7)

[Point 2.           Sample size calculation is fundamental for such a research.]

Although statistical power calculation is commonly used as a planning tool to determine the size of the study, there are several drawbacks of statistical significance testing; the major concern of such drawbacks include the temptation to dichotomize study results into qualitative categories, which notoriously have led to misinterpreting nonsignificant findings to be support for the null hypothesis (Bland JM, BMJ. 2012; Greenland S, Eur J Epidemiol. 2016, Am J Epidemiol. 2017; Rothman KJ, et al. Epidemiology. 2018). Study-size formulas are purely mathematical, which do not account for anything that is not included as a variable in the formula. In addition, focusing only on power may lead to neglect the most important study results: estimates and confidence intervals. In clinical trials, where the investigators are required to balance the value of greater precision in study results against the greater cost under the limited resources, advantages of calculating sample size may outweigh these drawbacks (Rothman KJ, et al. Modern Epidemiology 3rd edition, Philadelphia, PA. 2012). In our current case-control study, however, the primary purpose was not to focus on whether the P value falls below some cutoff, rather to evaluate our hypothesis whether ocular surface manifestations were differed by cataract surgical procedures in a relatively long-term, which were poorly understood due to the limited unilateral postoperative cases. We acknowledge the small sample size of this study and thus additional larger studies are expected. Nonetheless, to reduce random errors and increase the precision, we have strictly followed the standardized DED evaluation and all cases were evaluated by a single dry eye specialist; hence, we believe that our study design was feasible in this respect. In response to the reviewer’s comment, we have modified the DISCUSSION section as follows:

“First, due to the uniqueness of our study design, we had a small number of subjects who met the inclusion criteria and thus future studies in large sample size were required to confirm the current findings.”

(DISCUSSION, page 6)

[Point 3.           The Method is poor reliable/standardized. Did you measure the repeatability of results? Did you quantify the instilled fluorescein? Did you monitor the environmental conditions?]

We appreciate the reviewer’s comments. Details of our study protocol was described previously (Hanyuda A, et al. JCM 2019). Briefly, noninvasive assessment including fluorescein-assessment of tears and ocular surface epithelium was conducted by a single dry eye specialist (M.A.) under the current Japanese dry eye diagnostic criteria (Shimazaki J, et al. Atarashii Ganka 2007), to minimize the inter-investigator heterogeneity. In accordance with the standardized protocol for DED evaluation (Yokoi N, et al. Am J Ophthalmol. 2017), the fluorescein staining of tears was strictly monitored with no change of the subject’s aqueous tear volume, after putting 2 drops of saline solution with a fluorescein test strip (Showa Yakuhin Kako Co, Tokyo, Japan). The strip was touched gently to the central lower lid margin. We asked the patient to close the eye gently and briskly open the eye after several natural blinks. The investigator determined the starting point of eye opening as well as confirmed the reproducibility of tear break-up time (TBUT) by 3 successive observations and the mean value was used for the current analyses. All examinations were performed at a temperature of 18–25°C with humidity of 40-60%. In response to the reviewer’s comment, we have modified the METHODS section as follows:

“According to the standardized DED evaluation,[31] the fluorescein staining of tears was strictly monitored with no change of the subject’s aqueous tear volume, after putting 2 drops of saline solution with a fluorescein test strip (Showa Yakuhin Kako Co, Tokyo, Japan). The strip was touched gently to the central lower lid margin. We asked the patient to close the eye gently and briskly open the eye after several natural blinks. The investigator determined the starting point of eye opening as well as confirm the reproducibility of TBUT by 3 successive observations and the mean value was used for the current analyses. To minimize interexaminer heterogeneity, objective dry eye signs, including the mean TBUT and keratoconjunctival staining scores (0–9 points) based on the Japanese dry eye diagnostic criteria,[29] were all evaluated by a single dry eye specialist (M.A.). The corneal and conjunctival fluorescein staining scores were evaluated in three areas (the temporal bulbar conjunctiva, nasal bulbar conjunctiva, and cornea) and scored on a 0–3-point scale in each section (0: no damage, 3: damaged entirely). Then, each score was summed, and the maximum score possible was a total of 9 points.[29] All examinations were performed at a temperature of 18–25°C with humidity of 40-60%.”

(MATERIALS AND METHODS, page 8)

[The article is interesting, but the interpretation may be incorrect. In my opinion, the paper at the moment is not ready for publication.]

We appreciate the thoughtful suggestions and valuable insights by the reviewer. We believe that in the process of responding to the reviewers’ comments, our manuscript has substantially improved.

Reviewer 2 Report

The manuscript by Hanyuda et. al. tackles dry eye disease (DED) development after cataract surgery in unilateral cases, which has been poorly studied. Their cross-sectional study was conducted in 5 eye clinics with 89 unilateral pseudophakic subjects with 1+ year of follow-up after simple cataract surgery. The tear break-up times (TBUTs) and keratoconjunctival staining results were compared between phakic and pseudophakic eyes. The TBUTs were low and, therefore, abnormal. The same was true for the kerato-conjunctival staining scores. They conclude by stating corneal status was significantly worse in the pseudophakic eyes than in the contralateral phakic eyes, even after more than one year after implant surgery.

The study is significant for following unilateral cases of cataract surgeries and DED development, analysis after 1 year post-op, and a fairly large cohort.

The only minor criticism is that the references need to include more studies that are outside Asia (esp. choice of references 6-13), as it leads the reviewer to think the reference choice is biased. Surely, the authors can quote studies done in Europe, Australia or USA. 

Author Response

Hanyuda A et al.  biomedicines-711379R

Response to Reviewer 2 Comments

Thank you very much for your interest in our manuscript entitled “Persistently worsened tear break-up time and keratitis in unilateral pseudophakic eyes after a long postoperative period”. To aid in the re-review of this manuscript, we have included a point-by-point response to each comment. The reviewer’s comments are italicized and placed in square brackets. In addition, within the revised manuscript, we have used underlined text to highlight changes in response to the reviewers’ comments.

We appreciate the suggestions and comments by the reviewer. As a consequence of valuable suggestions, we believe that our manuscript has been much improved.

[Reviewer 2:  The manuscript by Hanyuda et. al. tackles dry eye disease (DED) development after cataract surgery in unilateral cases, which has been poorly studied. Their cross-sectional study was conducted in 5 eye clinics with 89 unilateral pseudophakic subjects with 1+ year of follow-up after simple cataract surgery. The tear break-up times (TBUTs) and keratoconjunctival staining results were compared between phakic and pseudophakic eyes. The TBUTs were low and, therefore, abnormal. The same was true for the kerato-conjunctival staining scores. They conclude by stating corneal status was significantly worse in the pseudophakic eyes than in the contralateral phakic eyes, even after more than one year after implant surgery.

The study is significant for following unilateral cases of cataract surgeries and DED development, analysis after 1 year post-op, and a fairly large cohort.]

We appreciate the reviewer’s complimentary comments.

[Point 1.           The only minor criticism is that the references need to include more studies that are outside Asia (esp. choice of references 6-13), as it leads the reviewer to think the reference choice is biased. Surely, the authors can quote studies done in Europe, Australia or USA.]

We appreciate the reviewer’s comments. As the reviewer suggested, we have included some studies conducted outside of Asia including USA and England (references 9 and 12).

Reviewer 3 Report

The manuscript collects a number of relevant data regarding the development of DED in patients after cataract surgery.

The aim is very interesting and the work is well structuredT, the experimental design is well organized, the results are shown in clear manner and the discussion is well articulate.

I suggest to accept it in the present form.

Author Response

Hanyuda A et al.  biomedicines-711379R

Response to Reviewer 3 Comments

Thank you very much for your interest in our manuscript entitled “Persistently worsened tear break-up time and keratitis in unilateral pseudophakic eyes after a long postoperative period”. To aid in the re-review of this manuscript, we have included a point-by-point response to each comment. The reviewer’s comments are italicized and placed in square brackets. In addition, within the revised manuscript, we have used underlined text to highlight changes in response to the reviewers’ comments.

We appreciate the suggestions and comments by the reviewer. As a consequence of valuable suggestions, we believe that our manuscript has been much improved.

[Reviewer 3:  The manuscript collects a number of relevant data regarding the development of DED in patients after cataract surgery.

The aim is very interesting and the work is well structuredT, the experimental design is well organized, the results are shown in clear manner and the discussion is well articulate. I suggest to accept it in the present form.]

We appreciate the reviewer’s complimentary comments.